# Study on the Micro-Abrasion Wear Behavior of PVD Hard Coating under Different SiC Abrasive Particles/Distilled Water Ratios

**DOI:** 10.3390/ma16082939

**Published:** 2023-04-07

**Authors:** Andresa Baptista, Gustavo F. Pinto, Vitor F. C. Sousa, Francisco J. G. Silva, Filipe Fernandes

**Affiliations:** 1ISEP—School of Engineering, Polytechnic of Porto, Rua Dr. António Bernardino de Almeida 431, 4200-072 Porto, Portugal; 2INEGI—Instituto de Ciência e Inovação em Engenharia Mecânica e Engenharia Industrial, Rua Dr. Roberto Frias 400, 4200-465 Porto, Portugal; 3CEMMPRE—Center for Mechanical Engineering Materials, Processes Department of Mechanical Engineering, University of Coimbra, Rua Luís Reis Santos, 3030-788 Coimbra, Portugal

**Keywords:** PVD, micro-abrasion, three-body abrasion, two-body abrasion, ball surface texture, TiN films, abrasive particles, silicon carbide particles

## Abstract

Microscale abrasion has been intensively used to study the wear behavior o several hard coatings, enabling the observation of different wear mechanisms. Recently, a study arguing whether the surface texture of the ball could influence the dynamics of abrasive particles throughout the contact was presented. In this work, the influence of the abrasive particles concentration able to change the texture of the ball was studied to understand its influence on the wear modes—rolling or grooving. Thus, tests were carried out using samples with a thin coating of TiN, deposited using the Physical Vapor Deposition (PVD) technique, and AISI 52100 steel balls etched over 60 s to induce a change in their texture and roughness were used. Three abrasive slurries were prepared with black silicon carbide (SiC) particles (average particle size of 4 μm) with different concentrations, 0.25, 0.35, and 0.45 g/cm^3^. The rotation speed used in the tests was 80 rpm and the normal loads applied in the study were 0.2 N and 0.5 N, and 1 N. After the wear tests, the coated samples and tracks on the surface of the balls were observed by SEM and 3D microscopy to understand the abrasive particle dynamics, evaluating the wear mode transition as well as the function of both applied load and slurry concentration. The tracks in the balls showed particles embedded on their surface. A lower concentration of abrasion was conducted to higher specific wear rate. Moreover, a predominant two-body wear mechanism was induced when the abrasive concentration was increased. There was an increase in the roughness of the scar and the surface of the balls with an increase in the abrasive particles’ concentration.

## 1. Introduction

Since the beginning of the development of hard coatings, there has been an absolute need to assess their wear behavior. The setup most used to characterize such properties has been the pin-on-disc [1,2], but other configurations have also been used [3]. However, some of these test typologies require very expensive equipment, which it is sometimes not very intuitive to use, and which requires samples with specific geometries [4]. Thus, it soon became necessary to develop a new, expeditious methodology for characterizing wear that could be associated with the continuous development of Physical Vapor Deposition (PVD) and Chemical Vapor Deposition (CVD) coatings [5], which have become increasingly common in most diverse applications, but with particular emphasis on cutting tools [6,7]. It was in this context that Rutherford and Hutchings [8,9] developed a new testing methodology based on micro-abrasion in the last years of the 20th century. This methodology is based on the principle of the controlled introduction of abrasive particles into a contact area, with those particles subjected to a normal force and creating a scar in the surface of the coated sample. These abrasive particles contained in a slurry are carefully provided to the contact area, and then forced to move across the contact area by a standardized ball animated by a rotational motion. This principle of test generates circular craters, which can drill (or not drill) the entire coating thickness. This process is often called the ball-cratering tester [10]. The principle is extremely simple, leading to the possibility of using relatively small and easily prepared samples, using reasonably inexpensive equipment, and presenting very good flexibility and repeatability. Furthermore, the handling of this type of equipment does not require a very refined specific training [11,12]. Given the advantages mentioned above, several groups of researchers dedicated their attention and efforts to the study of this methodology, with a view to its standardization. It was also necessary to analyze its limitations so that the test procedures and the analysis methodology were standardized, which was carried out by Gee et al. [13]. In order to find out whether the results generated by this new wear test configuration were reliable, Gee et al. [14] also conducted a specific work collecting results from 14 different laboratories working under the same conditions and on similar samples. The results obtained through microscopic analysis of the craters produced in these tests showed a dispersion of less than 2%, which made it possible to perceive that the reproducibility of the tests were good enough to validate the methodology. Nevertheless, the analysis of craters through microscopy was not unanimous among the researchers who were studying this methodology at that time. Effectively, Schiffmann et al. [15] reported additional difficulties in terms of accuracy in measuring the diameter of the craters when the coating was drilled and the substrate was hit. Thus, this group of researchers recommended the use of profilometry for the analysis of craters as the most appropriate, but, over time, microscopy has become the preference of researchers to measure the craters produced, as it allows them to jointly analyze the wear mechanisms that occurred during the test. Alternatively, Gee [16] suggested the use of a scanner to ensure a three-dimensional analysis of the craters generated during the tests, but it has almost never been preferred by other researchers, so there are a few works that have based their analysis on this methodology for analyzing craters. On the other hand, the main focus of the work of Kusano et al. [17] was to guarantee the dimensional accuracy in the measurement and the subsequent treatment of the obtained data, thereby establishing their own methodology. After a first phase of concern, which was with the way in which the results were obtained and the reproducibility of these results, attention was focused on the analysis of the effects of each parameter selected for the tests. The study carried out by Leroy et al. [18] made it possible to perceive for the first time that the size of the abrasive particles used in the test and their concentration are relevant in terms of influence to the type of wear mechanism. Furthermore, two well-known mechanisms in abrasion phenomena were identified: two-body abrasion, also known as grooving, and three-body abrasion, commonly known as rolling. These observations were also corroborated by Adachi et al. [19] and Cozza et al. [20,21]. Moreover, the latter ones confirmed that the level of load used in the micro-abrasion tests plays an important role in the wear mechanisms developed during the test, i.e., rolling or grooving. In a study conducted by Bose and Wood [22], these authors have suggested 0.3 N as the best option in terms of normal load to obtain a rolling steady-state behavior in micro-abrasion wear tests. However, other load levels have been used as well, which have attempted to analyze other related phenomena.

In experimental studies carried out by Leroy et al. [18], it was possible to verify that in the analysis of the wear in coatings of high hardness when there is perforation, there may be problems in the analysis of the craters. This problem is because there is a significant difference in behavior between the substrate, which is normally softer, and the coating crown, which remains resistant to the progression of the ball on the substrate. The conclusions shared initially by Leroy [18], Adachi et al. [19], and Cozza et al. [20,21] were later discussed by Cozza [23], who concluded that parameters such as the abrasive particle ratio in the slurry used as well as the normal load applied do not play a crucial role in the friction coefficient that is induced between the ball and the sample surface, and, thus, a direct correlation does not exist. Stachowiak et al. [24] investigated the influence of the abrasive particle size, reporting that larger abrasive particles induce a more severe wear rate on the sample surfaces. This happens because the track generated on the ball surface is progressing in width and morphology over the test, which tends to induce increased wear on the sample surface. This is even more effective when angular particles are used, promoting a non-linearity in wear rate evolution. Moreover, this non-linearity is even enhanced as the crater depth/width tends to increase, because a large number of particles tend to participate actively in the micro-abrasion process. It was also observed that this trend induces a gradual change from a rolling to a grooving wear mechanism. Given this observed transition, Shipway et al. [25] developed a model that allows the estimation of the dynamics of the particles in their movement between the ball and the counterface, taking into account the applied normal load and the way it is transmitted through the particles. Moreover, these authors state that special care is needed when understanding the simulation results obtained. In an attempt to understand whether the wear mechanism was maintained in the use of mono- or multi-layer coatings, Batista et al. [26] compared the wear behavior of a TiN monolayer coating with the TiN but one that consists of two layers. In this study, it was possible to observe that while the monolayer coating presented only grooving as a wear mechanism, the bilayer coating presented a mixed behavior, presenting both grooving and rolling. The effect of the abrasive particle size on wear mechanisms in ball-cratering tests was also the main focus of the work carried out by Andrade et al. [27], leaving the rest of the parameters unchanged. These authors found that larger abrasive particles have greater difficulty coming into the contact. However, when they do come into the contact, they tend to roll, inducing a mixed behavior that does not promote clear grooves in the crater. On the other hand, the smaller particles leave well-marked grooves in the center of the crater, making it clear that they are more easily embedded in the surface of the ball and that, with each pass, they highlight the grooves which were initially created. Thus, this work corroborated the crucial importance of particle size in defining the dominant wear mechanism in each test. On the other hand, Silva et al. [28] focused their attention on the material of the particles, studying three different materials as abrasives: diamond, SiC, and Al_2_O_3_. The size of the particles was kept as similar as possible. The craters generated were analyzed in detail, allowing the observation that the most uniform were those produced by the diamond abrasive, a phenomenon that was attributed to the fact that it is hardest material. Otherwise, the main concern of Ardila et al. [29] was to understand the effect of the ball material on the friction coefficient registered using the same coatings. It was concluded that harder ball materials induce increased friction coefficient, which, in turn, promotes fewer uniform craters and wrinkled grooves, but without a regular pitch. On the other hand, balls of softer materials showed a clear trend to embed the abrasive particles on their surface. In the first stage, the predominant wear mechanism is rolling but, over time, grooving becomes predominant. The equipment used in that work is a little bit different to the PLINT TE-66 usually used, presenting a test rig configuration. Pinto et al. [30] have extended this work, not centered on the friction coefficient, but on the wear mechanisms observed. The materials used in that work are also a little different, but present diverse hardness characteristics. A ball of AISI 304 stainless steel has been the most effective at dragging the abrasive particles, mainly due to its hardness level (between AISI 52100 steel and PTFE). Indeed, particles present more difficulty in being embedded on AISI 52100 steel, and PTFE allows particles to be completely embedded, inducing a less effective function in the contact. Regarding the wear mechanisms, grooving was predominantly observed when using AISI 52100 steel balls, while rolling was the prevalent mechanism in the tests carried out with balls made of AISI 304 stainless steel. However, the texture of the balls is also important in dragging out abrasive particles. Thus, Baptista et al. [31] promoted different etching times in balls initially provided with the same characteristics, inducing different surface topographies. Tests carried out with these balls and different normal loads showed that the more textured the balls are and the higher the normal load, the greater the wear rate. A similar study was carried out by Ardila et al. [32], allowing them to realize that ball topography varies during the tests, which induces an increased contact area and surface roughness, which becomes more effective with the dragging effect of the abrasive particles throughout the contact. Ball rotation speed effect was the focus of the work of Esteves et al. [33], showing a decrease in the wear rate as the rotation speed increases. The influence of the normal load on the abrasive particle dynamics was the main concern of Shipway and Hogg [34]. They also concluded that both the shape and the size of the abrasive particles play a crucial role on the wear rate, because particles provided with an irregular shape and a larger size present greater difficulties in penetrating into the contact. However, after entering, they promote lateral cracking, boosting the surface damage of the samples and increasing the wear rate. In turn, smaller particles enter more easily into the contact, the load is distributed by more particles, and the wear decreases. Regarding the ball surface roughness, this work is in line with the results reported by Andrade et al. [27]. Baig et al. [35] have confirmed that the abrasive particle size and the corresponding concentration in the slurry plays a crucial role on the wear behavior. The effect of the test temperature has also been investigated by Allsopp and Hutchings [36], proving that wear mechanisms observed at room temperature cannot be linearly translated to the wear mechanisms expected at elevated temperature.

Several hard coatings have been tested using a ball-cratering wear test configuration. Stack and Mathew [37] evaluated for the first time the wear behavior of WC/Co coatings, comparing them with uncoated stainless steel. The last one has presented the best wear performance of the WC/Co coating, mainly for elevated normal loads. Diamond and NCD (Nano Crystalline Coatings) coatings obtained by CVD were tested by micro-abrasion using diamond as the abrasive particles, which enabled the evaluation of the wear performance of these very hard coatings [38,39]. Rodríguez-Castro et al. [40] carried out a comparative study between uncoated CoCrMo substrates with CoB/Co2B coated ones by ball-cratering tests, mapping the conditions able to promote a change between grooving and rolling wear modes. Micro-abrasion has been also used by Cozza [41] to determine the friction coefficient of commercially used coatings, such as TiN and TiC, reporting that the friction coefficients computed are between 0.4 and 0.9. PVD TiAlN and TiAlSiN coatings have been investigated by Silva et al. [42] in terms of wear behavior, reporting that 5% Si in the composition of that coating is not sufficient to ensure a clear improvement of the wear resistance of that coating compared with the TiAlN initial coating. TiAlCrSiN hard coatings obtained by PVD have been also investigated by Martinho et al. [43] using micro-abrasion, comparing them with uncoated pre-treated tool steel usually used in molds for the injection of reinforced plastics. Authors have concluded that coating was responsible for an increase of 50% on the surface wear resistance, contributing in this way to a longer lifespan of molds devoted to the injection of plastics reinforced with 30% of glass fiber. With a view to solving a similar problem, Silva et al. [44] used a multilayer structure constituted by CrN/CrCN/DLC obtained by PVD, performing micro-abrasion tests and practical tests in an effective injection context, using coated inserts in an injection channel. The authors concluded that the results are not comparable, and that any gains achieved in micro-abrasion cannot be directly translated into benefits on the surface of the injection molds. Even around the same concern, Silva et al. [45] carried out a comparative study using different PVD coatings, concluding that TiAlN showed the best micro-abrasion wear performance but, as stated before, these results cannot be directly translated to the coating wear behavior in a real working context into the mold. Some investigators have used micro-abrasion tests to analyze the wear resistance of bulk materials. Stachowiak et al. [46] studied the wear behavior of high Cr content white cast irons in order to analyze which phases of their structure best resist wear. The study concluded that the carbides present in the white cast iron structure resisted wear much better, while the matrix was more easily worn. Moreira et al. [47] have also utilized micro-abrasion to investigate the advantages provided by the addition of WC-MMCs (Tungsten Carbide Metal Matrix Composites) into low carbon cast steels, achieving an increased wear resistance of 39% with the novel structure developed. Fernandes et al. [48,49] studied the influence of cast iron dilution on the abrasion resistance of Ni-based welds produced by a plasma transferred arc in order to use it as a hard facing coating to protect the surface of the molds that are used to produce glass bottles. The effect of an annealing treatment on the abrasion resistance of the welds was studied, too. A mapping of the wear mechanisms as a function of the slurry concentration and applied load was also studied by the authors for this type of hard facing coating in a separate paper [49]. Polymeric coatings [50] have been tested using micro-abrasion wear tests as well, showing very encouraging results.

The first PVD TiN hard coatings were deposited by the low-voltage electron beam evaporation in 1980 by Balzers company [51]. Despite the advances on the coating system, either by adding elements to TiN or by changing the architecture, configuration, or chemical composition of the coatings, TiN coatings produced by sputtering are still a quality standard in this field. Indeed, this coatings system is easy and simple to deposit, with adequate properties to protect the surface of specific parts against wear. Despite the tribological studies conducted on this coating system, using pin-on-disc and reciprocating sliding equipment’s abrasion, resistance of this material using microscale abrasion equipment was less explored. The current authors in a previous work studied the influence of the surface texture of the ball (20, 40, and 60 s etching) on the dragging of abrasive particles in micro-abrasion tests using different sizes of diamond abrasive powder particles in TiN films [31].

This study intends to extend the previous knowledge by investigating the micro-abrasion wear behavior of a TiN PVD hard coating using a different abrasive slurry with different concentrations and applied loads: 0.2 N, 0.5 N, and 1 N.

## 2. Materials and Methods

### 2.1. Material

#### 2.1.1. Substrate Material, Geometry and Balls Material

In this work, Uddeholm Calmax steel, chrome-molybdenum-vanadium alloy steel, (AISI D3; DIN X210Cr12; W.r 1.2080) was used as a substrate to deposit the TiN coating produced by sputtering. This substrate is in the annealed state with a hardness of 248 HB and Young’s modulus of 210 GPa [30,31]. The nominal chemical composition of the steel in wt% is: C—0.6%; Si—0.35%; Mn—0.8%; Cr—4.5%; Mo—0.5%; V = 0.2%.

The samples used were previously prepared with the following dimensions: 28 mm × 25 mm and 4 mm thick. They were then subjected to heat treatment following all the indications in the supplier’s technical sheet (F. Ramada S.A, Porto, Portugal), and DIN: X210Cr12, AISI D3, W.NR.: 1.2080, obtaining a final average hardness of 9.723 ± 0.044 GPa as a preparation prior to the deposition process of PVD thin films. After heat treatment, the surfaces were thoroughly cleaned in an ultrasound bath. This was followed by polishing the surface with 400, 600, 800, and 1200 grit sandpaper (crossing grinding directions from sandpaper to sandpaper), and then polishing it with diamond slurry until it had a mirror-like aspect. After this preparation, all the samples were coated with a coating of TiN produced by sputtering—PVD.

For the micro-abrasion tests, AISI 52100 steel balls were used. The 25.4 mm diameter balls were purchased in the polished state, and their surface morphology is shown in Figure 1a. They were subjected to a 15-min ultrasound cleaning to ensure the removal of oils and dirt from the supplier packaging conditions. However, the roughness was induced at the ball surface using 4% Nital solution for 60 s. The criterion for selecting this treatment and, therefore, this roughness lies in the results obtained in [30,31].

The surface state of the ball in the as-received condition and after chemical etching in 4% Nital is presented in Figure 1b. As expected, the as-received ball displays a smooth surface morphology, whilst the etched ball displayed a rough morphology. According to the roughness measurements, the roughness of the smooth ball is Sa = 0.047 ± 0.003 µm and the rough ball is Sa = 0.239 ± 0.009 µm. The EDS spectra acquired on the surface of the polished ball, representative of both balls’ surface conditions, is shown in Figure 1c. As expected, signals from iron (Fe) chromium (Cr) and silicon (Si) could be detected. Later, to better understand the influence of the roughness of the balls and their ability to retain or not retain abrasive particles, the tracks produced on the ball were also characterized by SEM.

#### 2.1.2. Abrasive Particles Used in the Micro-Scale Abrasion Tests

Silicon carbide powder SiC #1200, with 4 μm particle size average and 6 μm maximum value, according to the supplier, was used as an abrasive in the wear tests. The aspect of the particles observed in SEM is shown in Figure 2a, where it is observed that the particles have an irregular shape. According to Baig et al. [35], particle distribution influences the mechanisms developed in the micro-abrasion tests.

To confirm the size distribution of the particles, laser diffraction equipment—Malvern 2000 ParticleSizer Analyzer (Malvern Instruments Ltd., Malvern, UK) with a Hydro 2000G sample scattering unit––was used for the analysis of the particle size distribution. The mathematical models followed, considering the equipment, were the Mie or Fraunhoffer theory, in order to form and to generate a granulometric distribution based on the Equivalent Spherical Diameter Volume. Thus, the SiC abrasive samples were dispersed in water at 250 rpm and three readings were taken, using a size range of 0.020 to 2,000,000 µm and obscuration of 14.13%. There are usually three values to define abrasive powders in terms of micron size: D10, D50, and D90. These values represent, respectively, 10%, 50%, and 90% of the product below the named micron size. The Span parameter shows the width of the size distribution, allowing us to see how far apart the 10% and 90% points are, normalized to the midpoint. Thus, the volume-based size distribution range is defined as Span = (D90 − D10)/D50. Another important parameter is the mean volume moment (De Brouckere Mean Diameter). This reflects the size of the particles that make up most of the sample volume, being more sensitive to the presence of large particles in the size distribution. All distribution data were collected and analyzed following the Weighted Average of Volume D {4, 3}. The values obtained from the analyzed abrasive were as follows: SiC #1200: Span = (7.730 − 2.061)/3.510 = 1.615 µm, Weighted Average Volume D {4, 3} = 4.469 µm. The granulometry analysis, according to the Volume Weighted Mean D {4,3}, allowed a better understanding of the particle size dispersion, concluding that the size of abrasive particles according to the supplier (average of 4 μm and maximum value of 6 μm) is slightly below the values measured using the laser technique. The distribution of the particle size is shown in Figure 2b.

In conclusion, the laser technique used to measure and quantify the number of particles for each particle size allowed us to validate the average size of abrasive particles for the tests.

### 2.2. Methods

#### 2.2.1. Thin Film Coating Process

The reactor used for the deposition of the TiN coating on the samples was a CemeCon^®^ CC800/9ML DC magnetron sputtering machine with four Ti targets. Two gases were used in the process, namely, argon and nitrogen, with 99.99% purity. The homogeneity of the coating thickness was ensured by the rotation speed of the substrate. Table 1 presents the parameters used in the deposition process.

#### 2.2.2. Analysis of Morphology and Thickness of the Film

The preparation of the samples after deposition of the film through the PVD process was made in a sensitive way to avoid mechanical deformations of the substrate and the coating in the area where the samples were cut. Thus, the samples were partially cut on the back face using a disc saw. The samples were then immersed in liquid nitrogen for 30 min so that they could be easily broken by a hammer impact. The evaluation of the coating thickness was performed using an FEI Quanta 400FEG-SEM supplied with the EDAX Genesis X-ray spectroscope (EDS), which also allowed the characterization of the surface morphology and cross section morphology (FEI, Hillsboro, OR, USA). The SEM observations were conducted in secondary electrons (SE) and back scattered electrons detection (BSE) modes.

#### 2.2.3. X-ray Diffraction Analysis

To complement the characterization of the TiN-coated sample, X-ray diffraction equipment (Rigaku SmartLabSe diffractometer, Tokyo, Japan) was used to characterize the coating structure. The following parameters were used for the Spectrum acquisition: Cu Kα radiation with a wavelength of 1.5406 [Å]; Voltage: 40 [kV]; Step Size of 0.01 [°2Th.]; Scan Step Time of 1.0000 [s] and Current: 30 [mA].

#### 2.2.4. Adhesion Analysis

The adhesion of the film to the substrate was evaluated using two different setups: (i) the Rockwell-C indention tester and (ii) the scratch tester. The Rockwell-C indentations were performed in the EMCO M4U Universal Hardness Tester equipment (model M4U, EMCO-TEST Prüfmaschinen GmbH, Kellau, Kuchl, Austria) using a 1470 N load. Subsequently, all indentations were observed in an optical microscope (OM) OLYMPUS BX51M with a magnification of ×100 and ×200. The adhesion classification regarding the appearance cracks around the spherical depression was evaluated according to VDI 3198:1991 standard [52].

After this first analysis, scratch tests were performed by the scratch tester CSM REVETEST equipped with an acoustic emission detector. The scratch tests were conducted at a sliding speed of 10 mm/min at a load increase rate of 10 N/min from 0 to 30 N. The adhesion critical loads were evaluated according to the BS EN ISO 20502:2016 standard [53]. This type of test allows the quantification of normal load adhesion between the film and the steel substrate in the cohesive (Lc1) and adhesive (Lc2) failure modes. To increase the accuracy of the results, the procedure was repeated six times for each sample.

#### 2.2.5. Nano-Hardness Evaluation

A CSM Nanoindenter equipped with a properly calibrated Berkovich diamond indenter was used to analyze the hardness of the TiN film using the Oliver and Pharr method. The equipment used to perform the measurements was an NHTX S/N: 01-02934 device with the following parameters: acquisition rate: 10 Hz, maximum load: 150 mN, charging rate: 200 mN/min; unload rate: 200 mN/min and dwell time of 1 s.

#### 2.2.6. Roughness Analysis by AFM and 3D Optical Profilometer

To evaluate the surface roughness of the sample, an atomic force microscope (AFM), VEECO Multimode, was used. This equipment is equipped with a 7 nm radius probe and NanoScope 6.13 software. In the analysis, an area of 20 × 20 μm^2^ was evaluated, to reach the largest possible area, according to the samples. Three measurements in different zones of the coating were performed in Tapping Mode. For the evaluation of the roughness, the following parameters were considered: the arithmetic average roughness of the surface (Ra) and the maximum roughness (Rmax), according to EN ISO 4287.

Another method, less conventional than the AFM, was used to acquire topographic images and evaluate the roughness of the entire track on the balls as well as the wear craters, with nanometric resolution on the vertical axis. Thus, in this research, the topographic parameters were measured using the Sensofar S Neox 3D optical profilometer, using the Analysis SensoMap software 6.2. The surface parameters were calculated according to ISO 25178-2:2012 [54]. The following parameters were considered: the arithmetic mean of the surface roughness (Sa), and the maximum topographic surface height (Sz).

Data were obtained with a 10× interferometric objective (1754.40 × 1320.96 μm acquisition area) using the vertical scanning interferometry (VSI) technique in all craters, with a ×50 interferometric objective (350.88 × 264.19 [μm] acquisition area) in confocal mode (CO) inside the craters, and with a ×50 interferometric objective (1754.40 × 1320.96 [μm] acquisition area) in confocal mode (CO) on the tracks.

#### 2.2.7. Micro-Abrasion Test

The PLINT TE-66 micro-abrasion equipment was used for the wear tests. Three SiC abrasive slurry concentrations were prepared and used in the experiments. The weight of the abrasive mix in 100 mL of distilled water was: 25 g, 35 g, and 45 g. The ball rotation speed used on the experiments was 80 rpm, which corresponds to 0.105 m/s. The test duration was 500 cycles (corresponding to 39.90 m sliding distance), and three different loads 0.2 N, 0.5 N, and 1 N were used. Three tests were performed for each concentration with the respective load to ensure the reproducibility of the results. After the tests, all samples were kept in a desiccator at room temperature until their analysis in SEM. The humidity of the room was ~35% in all the tests.

After testing, all micro-abrasion craters were observed and measured using an optical microscope (OM) OLYMPUS (BX51M), and, subsequently, the wear mechanism observed in SEM. The specific wear rate was calculated applying Archard’s law [55].

## 3. Results and Discussion

### 3.1. Coatings Morphology and Thickness

The morphology of the TiN coating, as well as its thickness, can be seen in Figure 3. The film displayed a homogeneous surface morphology (see Figure 3a) where round features are visible, corresponding to the end of the columns. The cross-section morphology revealed that the film has a columnar morphology, where column growth since the substrate goes up to the top of the film (see Figure 3b,c). The coating is 3 µm thick. It should be noted that two levels of bias were used in the deposition process, those being −105 V and −90 V. This created two distinct zones on the film, as observed in Figure 3c. The EDS analysis carried out at the film confirmed the presence of Ti and N, as shown in Figure 3d.

### 3.2. Coating Structure

Figure 4 shows the X-ray diffraction pattern conducted at the film. The spectrum does not reveal significant differences in the structure as compared to other TiN coatings deposited by the same technique studied in the literature [30,31,56,57].

In the diffraction patterns, two type of phases can be identified: crystalline TiN and Fe, the last coming from signals of the substrate. The substrate can be indexed to the 2θ = 70.32° and 2θ = 93.48° angles.

Regarding the TiN signals, five peaks can be indexed. The three weakest peaks located at 2θ = 36.66°, 2θ = 74.07 and 2θ = 78.09° correspond to signals from the (111), (311) and (222) plans, respectively. The highest peaks of TiN are located at 2θ = 42.50° and 2θ = 61.66° corresponding to TiN (200) and TiN (220), respectively.

As already mentioned, the peaks and structure presented are in line with other investigations carried out, where the TiN peaks in the (111), (200), and (220) directions are the most common [30,31,56,57]. On the other hand, less common TiN peaks with less intensity correspond to the (222) and (311) directions [30].

### 3.3. Adhesion Evaluation

As described in Section 2.2.4, some indentations were initially made on the TiN samples at 1470 N following the VDI:3198:1991 standard [52] to obtain qualitative results on the adhesion of the film to the substrate. The result of the indentations can be seen in Figure 5a and in more detail in Figure 5b, where the presence of cracks is visible, marked with a red rectangle. The cracks identified on the indentation edge allow the classification of the cracking pattern as HF1 failure type, as reported in [42,58]. Thus, according to VDI Standard 3198: 1991 [52], this failure is classified as acceptable.

Scratch tests were also performed to evaluate the adhesion of the TiN film to the substrate. Figure 6 presents the scratch produced in the surface of the film. During the test, elastic acoustic waves were emitted, resulting from the release of energy at the time of the adhesion failures. A progressive load from 0 to 30 N was applied through an indenter, and an acoustic emission signal was emitted for a displacement of 10 mm.

Figure 6a shows the result of the acoustic emission signal of the test carried out. The first acoustic signals appear near to 18 N, corresponding to the first cohesive failure (Lc1) by lateral delamination, as can be seen in the image in Figure 6b. With the increase in the indentation load, the first adhesive failure type (Lc2) begins to appear close to 21 N, corresponding to the first exposure of the substrate, as seen in the marks in Figure 6c,d.

Crossing the images with the graph of the acoustic emission, it can be seen that the amplitude of the oscillations are slightly higher for adhesive failures and relatively smaller for cohesive failures. Similar to the Rockwell-C indentation tests, the scratch test suggests a good adhesion of the film to the substrate.

### 3.4. Micro-Hardness

The nanoindentation technique is particularly suitable for measuring the hardness of thin films whilst avoiding the influence of the substrate in the value obtained. Thus, a very low load was used and the indenter displacement was accurately controlled, which allowed the penetration depths to be accurately measured. Eight load levels were used in the test, namely: 5 nm, 10 nm, 20 nm, 50 nm, 75 nm, 100 nm, 125 nm, and 150 nm. For all measurements, a three-sided pyramidal Berkovich diamond indenter was used with a nominal edge radius of 20 nm that faces 65.3° ± 0.3° from the vertical axis, coupled to a properly calibrated nanoindenter (TTX–NHT, CSM Instruments).

The following hardness and Young’s modulus values were achieved: 18.14 GPa and 259.59 GPa, respectively. In comparison with other works, Young’s modulus presents a value 17% lower than that reported by [56] and 8% lower than [57].

Figure 7 shows the force-displacement curves resulting from a micro-hardness test that comprises three distinct phases: the first phase corresponds to the elastic deformation of the part being tested, the second phase corresponds to the elastoplastic deformation of the part under the indentation load, and the third phase corresponds to the unload.

Evaluating the obtained results, it is possible to observe that the test load penetrated 21.5% of the film thickness, and that there may have been a slight influence of the substrate, which justifies the values mentioned above. According to Martinho et al. [43,44], it is common to select an indentation depth lower than 10% of the film thickness to avoid the influence of the substrate contribution on the hardness and Young’s modulus.

Following the curve’s evolution with the increasing applied load, it can be observed that a constant slope up to the 50 mN load is observed. That slope becomes more accentuated above 55 mN when a load of 75 mN is applied, which may be a result of the influence of the substrate on the hardness reading. Since this slope is not very accentuated, it is possible to neglect the influence of the substrate on the measurements performed.

### 3.5. Roughness Results

The evaluation of the surface roughness of the sample was made through a topographic analysis by AFM. The method used allowed a more accurate assessment of the entire analyzed area, as seen in Figure 8. The scans carried out allow us to conclude that the surface of the coating is smooth and uniform. The method also allowed us to record roughness parameters, such as the arithmetic mean surface roughness (Ra = 26 ± 2 nm) and the maximum roughness (Rmax = 298 ± 9 nm). Comparing the roughness results with the ones obtained by other researchers, the value obtained here is slightly higher than the ones indicated by [24,28] and slight lower than the ones reported in [21,29,38,50].

The topography of the scars produced after microscale abrasion tests were analyzed by optical profilometry in the Sensofar S Neox 3D equipment, using the Analysis SensoMap software. All samples were first cleaned in an ultrasonic acetone bath for 5 minutes for topographic analysis. As already described, all surface parameters were calculated according to ISO 25,178-2:2012. The topographical patterns collected were restored using the software mentioned above, since they had some surface oxidation, and also to avoid reflections.

Three images were selected for a load of 0.2 N to demonstrate the evolution of the craters’ surface roughness, as representative for the other loads tested with an increasing abrasive slurry concentration, since the evolution was similar (see Figure 9a,c,e). The increase in load also induced an increase in roughness, as depicted in Table 2.

In the same images, it is possible to identify the wear resistance of the film through the crater contour, where the light blue is the abrasive inlet and the darker blue is the counter outlets with greater wear. The counter-outputs show that for all concentrations with which the abrasive particles came into contact, many of them stayed in the grooves created by the three-body abrasion (grooving), and those that are aggregated in the balls remain in contact.

Figure 9b,d,f also show that in all tests, the substrate was exposed, since the depth of the scar is higher than the film thickness and the substrate has suffered wear. Table 2 summarizes the roughness measurements acquired at the scar surface for the different loads and abrasive concentrations. Additionally, for better perception, the roughness values were plotted in Figure 10 to better observe the tendency of the roughness evolution. From Table 2 and Figure 10, it is possible to state that there is a tendency to increase the surface roughness of the craters as both the normal load applied and the abrasive slurry concentration are increased. A similar observation was made by Pinto et al. [58], which studied the different wear mechanism using alumina abrasive particles, textured balls, and increasing sliding distance.

### 3.6. Micro-Abrasion Analysis

Table 3 presents the mean crater diameters, the wear volume, the specific wear rate, and the respective standard deviations of the scars produced by micro-scale abrasion equipment for all loads and concentrations of the studied abrasive slurry.

Following Archard’s law, the specific wear rate was calculated for the different abrasive slurry concentrations (see Figure 11). As can be observed, increasing the abrasive slurry concentration progressively decreases the specific wear rate. This is explained by the changes on the wear mechanisms in the open scars, as explained later.

Looking at Figure 12 for a fixed abrasive slurry concentration, it can be observed that an increase in the diameter of the craters and, consequently, an increase on the volume of the scars as the load increases from 0.2 up to 1 N. Therefore, the increase in load caused greater wear on the craters, which is in line with what was described in micro-abrasive wear tests with a rotating ball [23]. However, by fixing the load and varying the abrasive slurry concentration, a decrease in the size of the craters was observed with increasing concentrations. Still, this decrease is not very significant, and, as will be explained later, this is caused by changes in the wear mechanisms on the scars produced by the tests.

The scars produced by the micro-abrasion setup were characterized by SEM to identify the dominant wear mechanisms. Figure 13a–c are the images collected in SEM for a load of 0.2 N and for the abrasive slurry concentrations of 0.25 g/cm^3^, 0.35 g/cm^3^, and 0.45 g/cm^3^, respectively. Figure 13d–f are the images taken in SEM for a load of 1 N and for the abrasive slurry concentrations of 0.25 g/cm^3^, 0.35 g/cm^3^, and 0.45 g/cm^3^, respectively. The images analyzed in SEM for the load of 0.5 N showed the same trend. In Figure 13, it the perforation of the coating and the apparent oxidation of the surface are both evident, which were confirmed by EDS (see Figure 14b,c). Additionally, independently of the load, it can be observed that the craters produced with the lower abrasive slurry concentration displayed a mix of a rolling/grooving wear mechanism. Increasing the abrasive concentration slurry promotes the formation of a grooving wear. This is an unexpected result, according to the literature, as increasing the abrasive slurry concentration should favor the rolling wear mechanism [49]. Such a difference can be explained by the roughness/surface condition of the ball, as will be seen later.

Figure 14 presents a more detailed analysis of the crater center with a load of 0.5 N for a slurry concentration of 0.45 g/cm^3^. Figure 14a shows the two-body wear mechanism (grooving) where it was possible to identify SiC abrasive particles embedded in the grooves (see Figure 14e,f). It should be noted that between the grooves, a deposition of abrasive slurry was observed, where it was not possible to measure the grain size since it appeared as a zone of accumulation of SiC slurry, as confirmed by the EDS analysis shown in Figure 14d. It is also important to highlight the existence of oxidation on the surface of the craters, as revealed by EDS analyses in Figure 14c. Another particularity was observed after cleaning the specimens with ultrasounds for SEM observations: some oxidized particles released from the surface, forming a lighter zone identified as zone Z1 in Figure 14a, and confirmed by EDS Figure 14b.

Considering the specific wear and the wear mechanisms presented above, it can be affirmed that the change from mix mode wear (rolling + grooving) to pure grooving explains the decrease in the specific wear rate with increasing abrasive slurry concentration, since grooving is well known to be a mechanism less efficient in removing materials compared to rolling. Contrary to the literature, where increasing the abrasive slurry concentration favored rolling behavior (for instance, Trezona et al. [56]), the grooving wear mechanism established in the current study that the increasing slurry concentration can be explained by the roughness induced on the ball surface by Nital etching. Indeed, due to the increase in abrasive slurry concentration, particles are able to be aggregated easily in the grooves of the surface roughness of the ball, causing more scratching and grooves in the craters. This trend was also observed in another study using the same type of TiN film with balls etched with 20, 40, and 60 s of 4% Nital, for other types of abrasive, and for diamond with different particles size: powder 1–2 μm, powder 2–4 μm, and powder 4–6 μm [31].

### 3.7. Balls Analysis

After wear tests, it was possible to observe changes on the surface condition of the balls. Since the trend of the texture of the tracks was the same for all loads, Figure 15a–c shows the topography of the ball tracks for 1 N load and abrasive particle concentrations of 0.25 g/cm^3^, 0.35 g/cm^3^, and 0.45 g/cm^3^, respectively, which are representative of the other testing conditions. From Figure 16a, it is possible to observe that the higher the abrasive concentration, the higher the number of particles embedded on the ball surface. According to the EDS analysis shown in Figure 16b–d, those particles correspond to SiC particles embedded on the surface of the ball. The increase in the number of particles embedded on the ball justifies the change from mix a rolling/grooving wear to a grooving wear. Indeed, according to the literature, pure grooving wear is less effective in removing material as compared to rolling or a mixture between rolling and grooving wear modes. Thus, a decrease in the specific wear rate is expected with the increase in abrasive particle concentration.

Even for the highest applied load (1 N), which would produce a higher wear of the ball, as suggested by the higher scars produced on the film, it is possible to observe that the roughness of the surface of the ball was not deeply modified. This allowed the SiC particles to be continuously accommodated in the grooves of the balls, promoting grooving wear.

The results of the roughness measurement by 3D profilometry in the center of the ball tracks is presented in Table 4, with the respective standard deviations. The results show a progressive increase in roughness with the increase in the applied normal load and the increase in the concentration of the SiC abrasive slurry. This corroborates the production of a purer abrasion wear mechanism due to the easier incorporation of abrasive particles on the ball, as explained previously.

To help understand what is happening on the tracks, a 3D scan was performed. Figure 17a,b shows two images of the tracks with a load of 0.2 N and concentrations of 0.25 g/cm^3^ to 0.45 g/cm^3^, respectively. Looking at Figure 17a,b, the trailhead for the lower concentration is wider, which resulted in the slightly larger craters observed by SEM.

## 4. Conclusions

The main goal of this work was to study the micro-scale abrasion resistance of a TiN film using different concentrations of SiC abrasive and different loads.

TiN coating was selected, since this type of coating is still a high-quality standard coating for different mechanical applications and their abrasion resistance is little studied.

Based on the observations made, it was possible to conclude the following:The micro-abrasion study showed that increasing the load and keeping the abrasive concentration constant led to an increase in the diameter and volume of the craters, promoting the formation of grooving wear. However, by varying the concentration and fixing the load, there was a slight decrease in the diameter and volume of the craters.Analysis by 3D profilometry performed on the surface of the craters revealed an increase in roughness with increasing abrasive slurry concentration and applied load, since the particles can be easily aggregated in the grooves of the surface roughness of the ball, causing more scratches and grooves in craters.Applying Archard’s Law, it was observed that the specific wear is slight lower for higher concentrations of abrasive slurry. This was attributed to the change of mix rolling/abrasion wear to pure grooving wear in the scars.Regarding the balls used, it was proved by SEM that the roughness of the ball prevailed, as the tests conducted against different loads and different abrasive concentrations. Aggregated SiC abrasive particles were detected in the tracks of all balls, and the test situation where this aggregation was most felt was at a concentration of 0.45 g/cm^3^ and a load of 1 N. Thus, it was proved that the attack on the ball ensured the aggregation of particles, promoting the mechanisms of wear to abrasion, and even after the tests, the particles remained embedded in the roughness of the balls after the test.The 3D profilometry evaluation of the ball tracks showed that increasing the concentration of abrasive particles and keeping the load constant significantly increases the roughness as well as the contact area, which was reflected in a wider wear track. Taking into account the tracks of the balls, a greater presence of particles was detected for the 1 N load and a concentration of 0.45 g/cm^3^. The 3D analysis allowed us to corroborate this effect, presenting the highest roughness value, Sa = 0.664 ± 0.008 μm, in the tests carried out.

## Figures and Tables

**Figure 1 materials-16-02939-f001:**
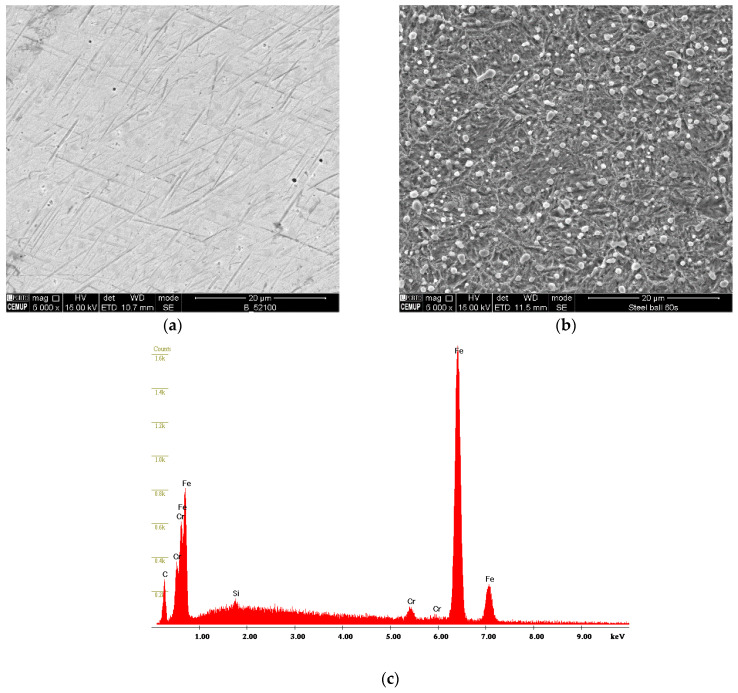
Ball surface morphology of: (**a**) as received ball, (**b**) etched ball in a 4% Nital solution for 60 s, and (**c**) EDS spectra of the polished ball.

**Figure 2 materials-16-02939-f002:**
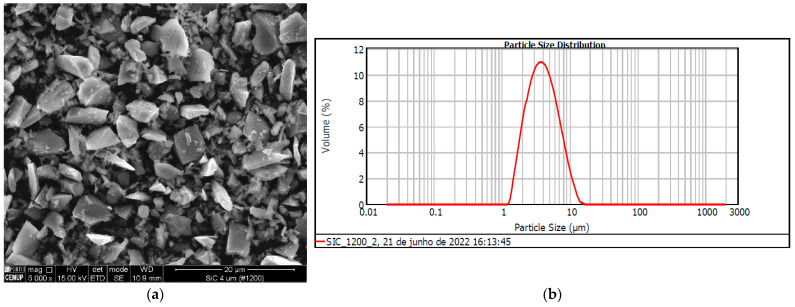
(**a**) SiC abrasive particles aspect, (**b**) distribution of the particle’s size.

**Figure 3 materials-16-02939-f003:**
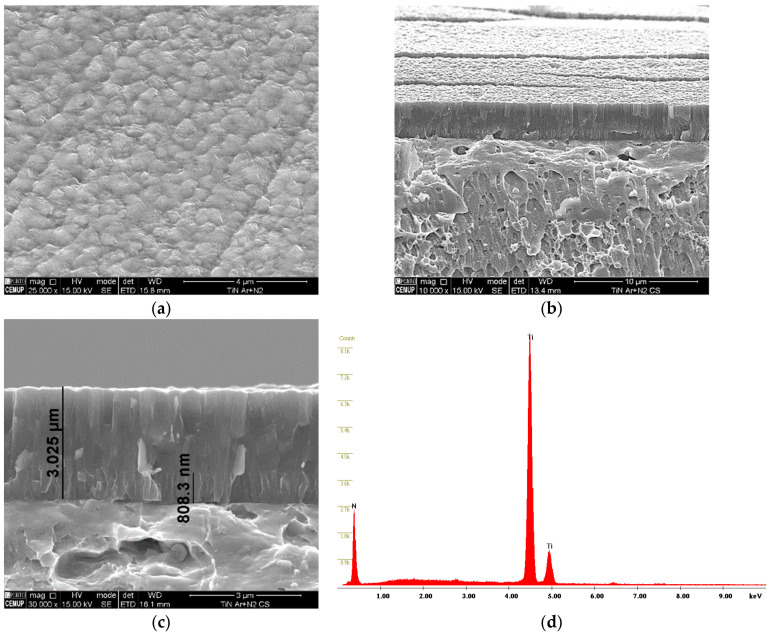
Morphological characterization of the TiN coating: (**a**) coating surface, (**b**) coating surface and thin film cross-section, (**c**) thin film cross-section, (**d**) TiN EDS spectra.

**Figure 4 materials-16-02939-f004:**
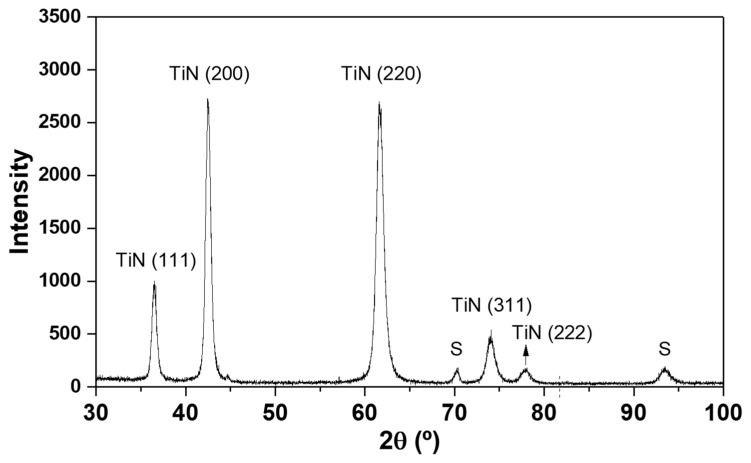
XRD diffraction patterns of the TiN thin film deposited by PVD.

**Figure 5 materials-16-02939-f005:**
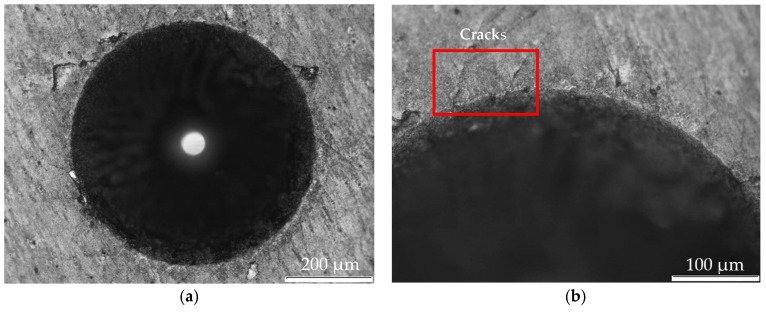
Indentation performed on the surface of the coatings using an optical microscope: (**a**) magnification of ×100, (**b**) a magnification ×200.

**Figure 6 materials-16-02939-f006:**
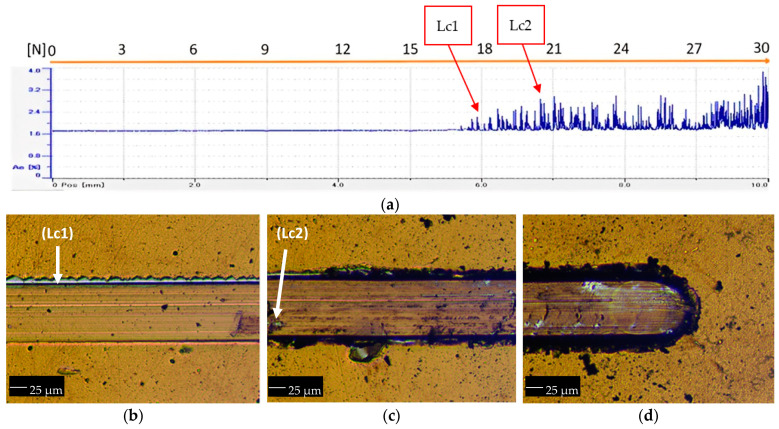
Scratch test progressive load (0–30 N) of TiN film: (**a**) acoustic emission signal as a function of scratch distance, (**b**) cohesive failure by lateral delamination (Lc1), (**c**) adhesive failures by the first exposure of the substrate (Lc2), and (**d**) area of the scratch end.

**Figure 7 materials-16-02939-f007:**
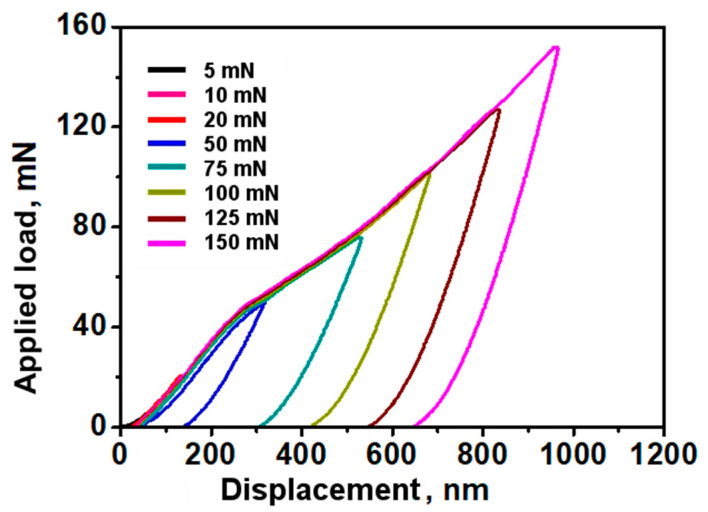
Load–displacement curves of indentations made on the TiN film.

**Figure 8 materials-16-02939-f008:**
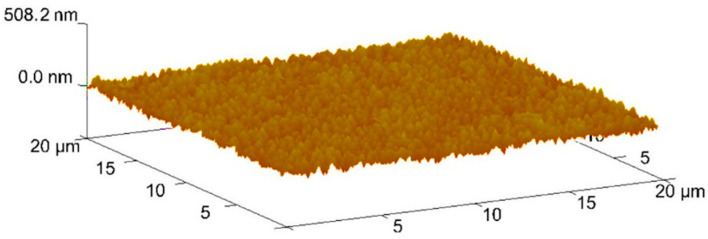
Load Topography of the coated surface analyzed by AFM.

**Figure 9 materials-16-02939-f009:**
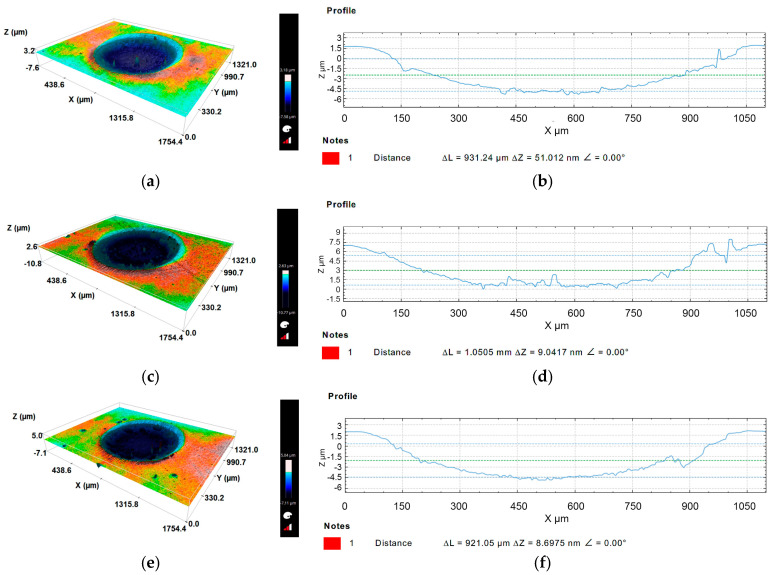
Crater surface topography analyzed by 3D profilometry for a load of 0.20 N: (**a**) slurry concentration (SC) 0.25 g/cm^3^; (**b**) detail of roughness in profile (SC) 0.25 g/cm^3^; (**c**) slurry concentration 0.35 g/cm^3^; (**d**) detail of roughness in profile (SC) 0.35 g/cm^3^; (**e**) slurry concentration 0.45 g/cm^3^; (**f**) detail of roughness in profile (SC) 0.45 g/cm^3^.

**Figure 10 materials-16-02939-f010:**
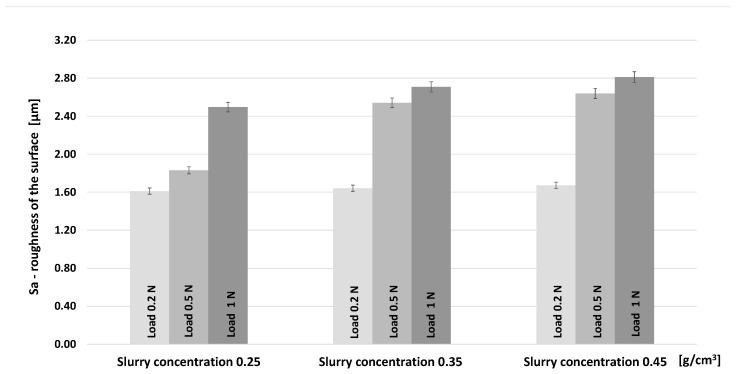
Roughness parameter trends (Sa) of the scars resulting from the micro-scale abrasion tests for the different abrasive slurry concentrations and different loads (0.2 N, 0.5 N and 1N).

**Figure 11 materials-16-02939-f011:**
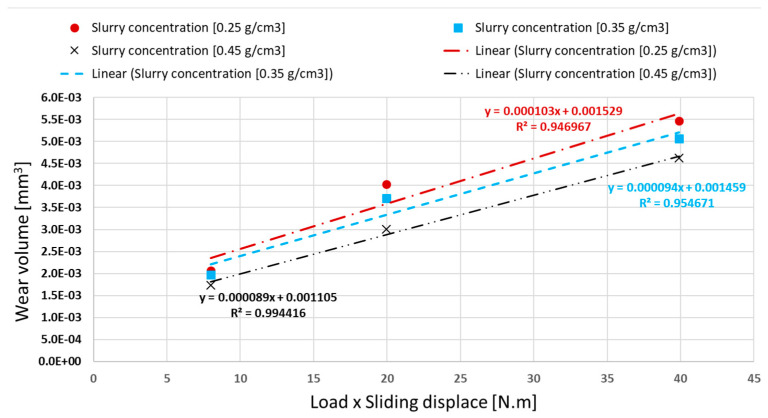
Results of application of Archard’s linear equation for specific wear rate for the test conditions.

**Figure 12 materials-16-02939-f012:**
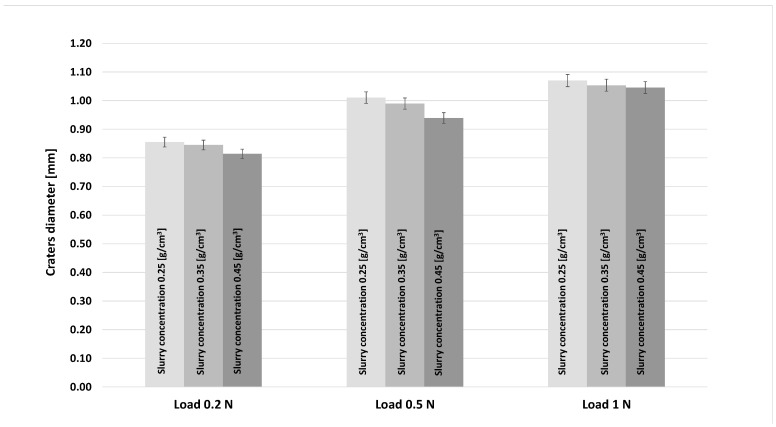
Roughness parameter trends (Sa) for different abrasive mass loads such as 0.2 N, 0.5 N and 1N.

**Figure 13 materials-16-02939-f013:**
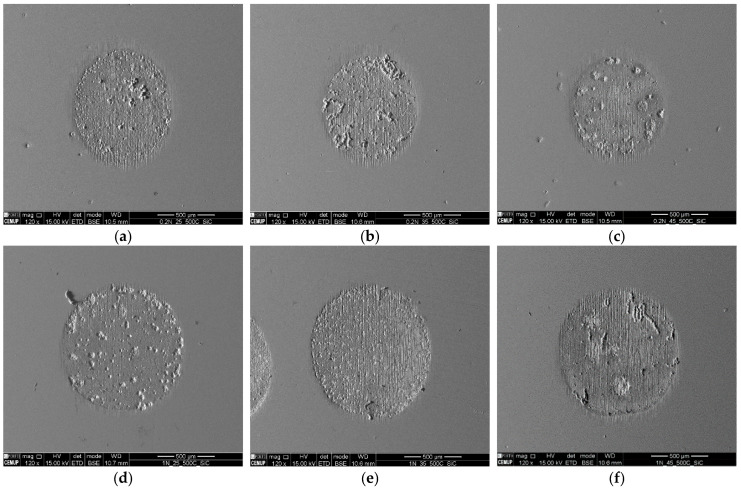
SEM images of the craters: (**a**) Load 0.2 N, (SC) 0.25 g/cm^3^; (**b**) Load 0.2 N (SC) 0.35 g/cm^3^; (**c**) Load 0.2 N (SC) 0.45 g/cm^3^; (**d**) Load 1N, (SC) 0.25 g/cm^3^; (**e**) Load 1 N (SC) 0.35 g/cm^3^; and (**f**) Load 1 N (SC) 0.45 g/cm^3^.

**Figure 14 materials-16-02939-f014:**
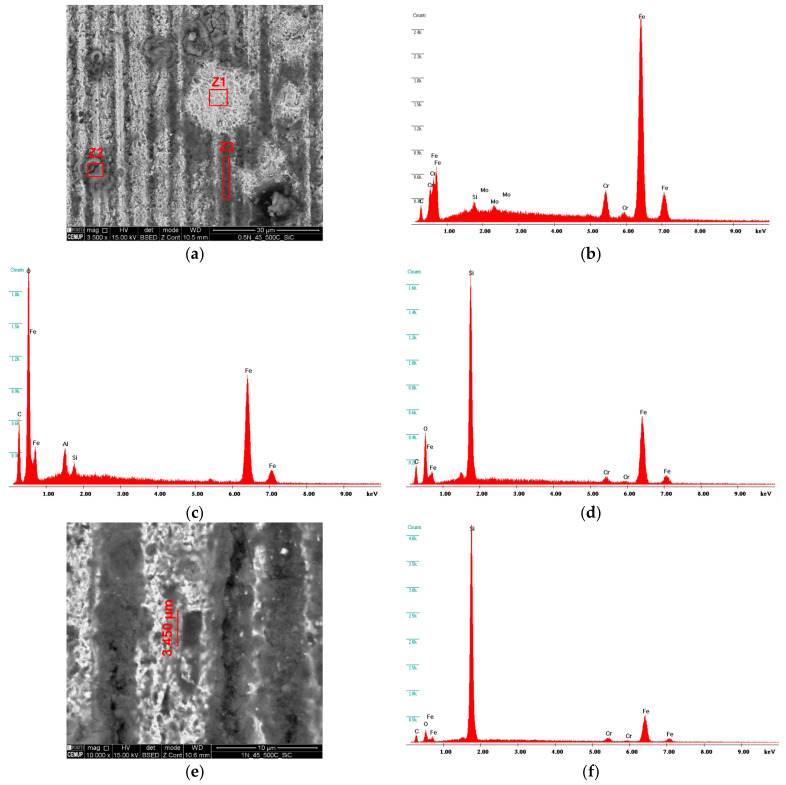
SEM analysis in craters subjected to a load of 0.5 N and SC of 0.45 g/cm^3^: (**a**) identification of the presence of SiC particles and oxidation; (**b**) EDS spectrum in the Z1 zone indicating oxidation release; (**c**) EDS spectrum in the Z2 zone indicating oxidation; and (**d**) EDS spectrum in the Z3 zone indicating deposited SiC Particles; (**e**) groove with aggregated SiC particle; (**f**) EDS spectrum indicating a SiC particle with 3.450 µm.

**Figure 15 materials-16-02939-f015:**
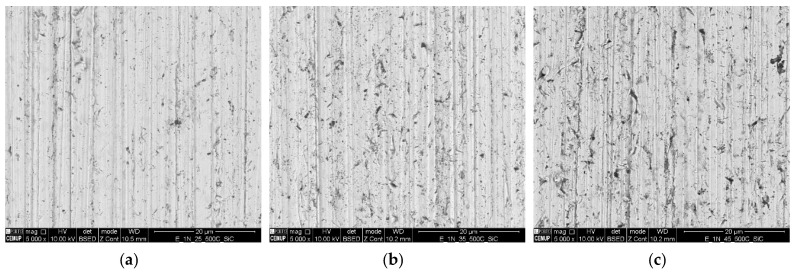
Analysis of the center of the ball tracks by SEM for abrasive SiC with a normal load of 1 N for ball etched 60 s: (**a**) (SC) 0.25 g/cm^3^; (**b**) (SC) 0.35 g/cm^3^; and (**c**) (SC) 0.45 g/cm^3^.

**Figure 16 materials-16-02939-f016:**
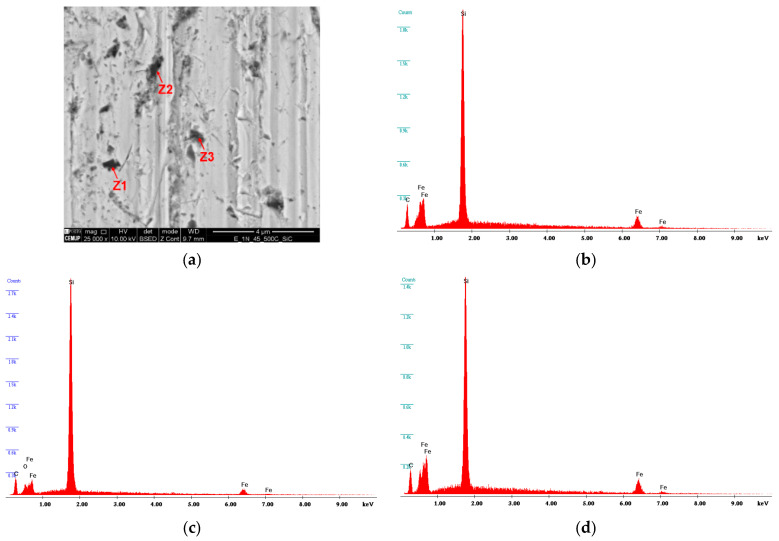
Analysis of aggregation of abrasive particles in the center of the rail for SC 0.45 g/cm^3^ with normal load of 1 N: (**a**) SEM analysis in the inside of the trail; (**b**) EDS spectrum in the Z1 zone indicating SiC Particle; (**c**) EDS spectrum in the Z2 zone indicating SiC Particle; and (**d**) EDS spectrum in the Z3 zone, indicating SiC Particle.

**Figure 17 materials-16-02939-f017:**
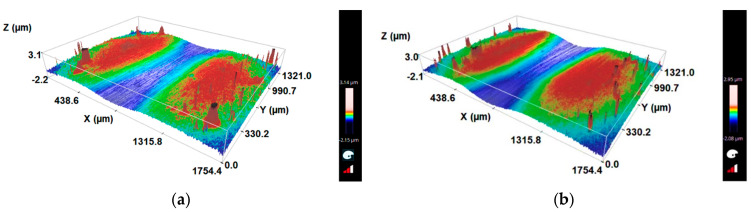
Topography of the ball tracks analyzed by 3D profilometry for: (**a**) slurry concentration (SC) 0.25 g/cm^3^ and load 0.2 N and (**b**) (SC) 0.45 g/cm^3^ and load 0.2 N. Note that the xx axis in the figures are different, thus the track of (**a**) is wider than (**b**).

**Table 1 materials-16-02939-t001:** PVD deposition parameters.

Deposition Parameters	Value
Gas pressure [Pa]	0.650
Temperature [K]	753.15
Target current [A]	10
Bias voltage [V]	−105 to −90
Deposition time [min]	123
Chamber gases	Ar 330 sccm, N_2_ 75 sccm
Rotation speed of the substrate holder [rpm]	1

**Table 2 materials-16-02939-t002:** Results obtained from the amplitude parameters (Sa and Sz) in the height deviation of the topography of the craters. Surface texture ISO 25178; Height parameters (μm).

Slurry Concentration [g/cm^3^]	0.25			0.35			0.45		
Load [N]	0.2	0.5	1	0.2	0.5	1	0.2	0.5	1
Sa [μm]	1.611 ± 0.104	1.830 ± 0.112	2.496 ± 0.146	1.641 ± 0.103	2.540 ± 0.149	2.707 ± 0.175	1.672 ± 0.110	2.638 ± 0.151	2.811 ± 0.180
Sz [μm]	12.15 ± 0.68	14.66 ± 0.69	18.27 ± 0.72	13.40 ± 0.69	17.67 ± 0.72	14.96 ± 0.70	10.76 ± 0.68	15.48 ± 0.71	14.34 ± 0.67

**Table 3 materials-16-02939-t003:** Mean crater diameters; wear volume, specific wear rate of the scars produced by micro-scale abrasion equipment as a function of the loads and concentrations of abrasive slurry used.

Slurry Concentration [g/cm^3^]	0.25			0.35			0.45		
Load [N]	0.2	0.5	1	0.2	0.5	1	0.2	0.5	1
Ø medium [mm]	0.855 ± 0.010	1.010 ± 0.009	1.070 ± 0.010	0.845 ± 0.021	0.990 ± 0.017	1.054 ± 0.014	0.814 ± 0.015	0.939 ± 0.019	1.046 ± 0.018
V—Wear volume [mm^3^]	0.00207	0.00403	0.00507	0.00197	0.00371	0.00477	0.00170	0.00301	0.00462
K—Specific wear rate [mm^3^/N.m]	0.000103			0.000094			0.000089		

**Table 4 materials-16-02939-t004:** Results obtained from the amplitude parameters (Sa and Sz) in the height deviation of the topography of the balls. Surface texture ISO 25178; Height parameters (μm).

Slurry Concentration [g/cm^3^]	0.25			0.35			0.45		
Load [N]	0.2	0.5	1	0.2	0.5	1	0.2	0.5	1
Sa [μm]	0.329 ± 0.002	0.437 ± 0.006	0.591 ± 0.008	0.479 ± 0.006	0.560 ± 0.007	0.614 ± 0.009	0.555 ± 0.007	0.605 ± 0.008	0.664 ± 0.008
Sz [μm]	8.665 ± 0.452	9.175 ± 0.510	14.282 ± 0.678	7.244 ± 0.359	12.947 ± 0.679	10.321 ± 0.601	26.201 ± 0.845	10.765 ± 0.668	10.344 ± 0.612

## Data Availability

Not applicable.

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
