# Peer review of "Study on the Micro-Abrasion Wear Behavior of PVD Hard Coating under Different SiC Abrasive Particles/Distilled Water Ratios"

_materials, 2023, doi:10.3390/ma16082939_

Round 1

Reviewer 1 Report

This paper investigated the influence of SiC abrasive concentration and load on the micro-scale wear performance of TiN coatings. The experimental results demonstrated that both SiC abrasive concentration and load significantly affect the wear behavior of TiN coatings, especially under high load and high abrasive concentration conditions, where the wear rate of TiN coatings increases noticeably. The authors conducted an in-depth analysis of the experimental results and propose some meaningful conclusions. However, there were several issues that need to be addressed:

 1.The paper contains several incorrect references to images, which should be revised.

 2.The experimental section lacked detailed method descriptions and operational steps, particularly in terms of sample preparation and testing processes, which may result in poor data uncertainty and repeatability. Therefore, it was recommended to add a detailed description of the experimental methods, including sample preparation and testing conditions.

 3.The interpretation of the experimental results was overly simplified, lacking in-depth analysis and discussion of the data. The paper needed to provide a more thorough analysis and discussion of the experimental results, highlighting their significance and importance, and proposing possible mechanisms and explanations.

 4.Some experimental details and methods, such as coating preparation methods, sample size and shape, and experimental temperature and humidity, were not fully described in the paper. Therefore, it was suggested that these be supplemented and described in detail.

 5.The language expression in the paper needed further refinement. Some sentences were unclear or contain grammatical errors, which should be revised and polished.

Author Response

The answers are attached in a word file.
Best regards

Reviewer 2 Report

The current paper is about microabrasion wear of PVD hard coating under different SiC abrasive particles concentrations and loading conditions.

In general, the paper is well written and the experimental work is considerable. Nevertheless the lack of details for some results and the lack of analysis/explanations on the wear behavior don't really highlight the paper value. The main conclusion on the change of mix rolling/abrasion wear to pure grooving wear is not enough emphasized, even in the conclusion lost in the middle of global observations.

Global remark: Pictures from SEM, EDS, ... in the different figures are usually low quality or with too small legends, annotations, axis values...etc (eg Fig. 7, Fig 10 very bad quality)

- l. 17: PVD or sputtered is missing in the abstract.
- l. 39: PVD and CVD are not defined
- l. 234: Missing Ref for substrate properties?
- l. 237: "...heat treatment following all the indications in the supplier's technical sheet" --> Too vague and no reference
- l. 238: Hardness increasing from 1.8 to 9.7GPa ? How characterized? By nanoindentation? Mechanisms?
- l. 247: NITAL or Nital, please use the same form along the article
- Fig. 1c: Please explain the "white particles" at the surface of the etched sphere. Is it roughness or contamination? Any EDS of the etched sphere?
- l. 269: Not sure that it is relevant to give supplier values and to look for confirmation. This paragraph should focus on the particle size characterization and obtained values, as we have no clue about the relevance and the quality of providers values. Reference to provide in the case the authors want to keep providers values for comparison.
- l. 287: Please give average + STD values for span and D instead of max value.
- Table 1: Symbol of Kelvin unit is K not k, and what is mln?
- l. 331, 342, 354: Why not giving references for the VDI standard, O&P method, EN ISO 4287?
- l. 364: vertical scanning instead of vertical displacement?
- Figure 7: Please position LC1, LC2 anc LC3 on the AE graph
- l. 456: At which depth the authors have obtained these values? Is it before the change of slope?
- l. 460: I don't see the 3 phases, creep? Maybe if the authors plot load or displacement as a function of time we could see some creep? Is it creep or thermal drift that is characterized?
- l. 467: Have you consider any densification effect for this slope change? Anything in the literature?
- Table 2: Please explain why Sz is not following exactly the same trend as Sa for 0.35 and 0.45g/cm3? What do you mean by "Surface IS O 25178 / Height (μm)" in the table title?

- l. 597: Ref  to support this assertion?
- Figure 12: Please provide the STD if possible and the regression coefficient for each lines? And y-axis is Wear Volume not Volume Wear
- l. 638: "it is clear", sorry but not for me. Please quantify this observation. For me Fig 18a and 18b are almost the same.
- Table 4: Same question as for Table 2

Author Response

(The authors gave the same response as above.)

Reviewer 3 Report

This paper is relatively well written and also informative to the research community of surface modification through coatings and further wear analysis, however, some points need to be further clear before publishing:

[1].  Kindly be clear in the abstract whether Diamond abrasive particles or SiC abrasive particles were used while testing.

[2].  The last paragraph of the introduction which usually highlights the summary of executed work and the novelty of the presented work is quite weak and needs to modify.

[3].  Fig. 3 is not appropriate, remove the unnecessary items in the background. Also, label the components of the equipment properly.

[4].  The identified cracks must be highlighted with marking in Fig. 6(a,b).

[5].  How you have executed the scratch test, also attached the equipment picture here.

[6].  Adjust Fig. 7 along with the caption.

[7].  How do you assess the wear volume, kindly describe the working and mechanism.

[8].  The conclusion needs to be revised. More specific details that are observed need to be quantitatively written in this section while the generalized trending in the abstract. 

Author Response

(The authors gave the same response as above.)

Round 2

Reviewer 1 Report

The authors have revised the manuscript according to the comments raised.

Author Response

The authors thank the comments and suggestions, very useful for improving the article.
An extensive improvement of the English Language and Style was made.
Best Regards

Reviewer 2 Report

Thanks for the update. There is just a minor error to solve Fig.6 with 2X Lc2 given on the different graphics.

Author Response

The authors thank the comments and suggestions, very useful for improving the article.
Figure 6 was rectified as suggested.
An extensive improvement of the English Language and Style was made.
Best Regards